# EFFECT OF DOCUMENT PACKING ON THE LATENT MULTI-HOP REASONING CAPABILITIES OF LARGE LANGUAGE MODELS

## ABSTRACT

The standard practice for training large language models involves packing multiple documents together to optimize computational efficiency. However, the impact of this process on the models' capabilities remains largely unexplored. To address this gap, we investigate how different document-packing strategies influence the latent multi-hop reasoning abilities of LLMs. Our findings indicate that packing can improve model performance compared to training on individual documents, at the expense of more compute. To further understand the underlying mechanisms, we conduct an ablation study, identifying key factors that explain the advantages of packing. Ultimately, our research deepens the understanding of LLM training dynamics and provides practical insights for optimizing model development.

## 1 INTRODUCTION

Document packing is a common technique in LLM training that optimizes computational resources (Raffel et al., 2019; Brown et al., 2020; Rae et al., 2021; Chowdhery et al., 2022; Gunasekar et al., 2023). Instead of padding each document in a batch to match the longest one, document packing concatenates multiple documents within a sequence. This minimizes padding and enhances training efficiency.

Despite its widespread use, the fundamental impact of this approach on LLM training dynamics and performance remains underexplored. Strategies like packing related documents (Shi et al., 2023) and reducing truncation (Ding et al., 2024) have shown improvements in downstream task performance. Analogously, research has indicated that document segmentation can impair LLMs' ability to recall information effectively (Prato et al., 2023) and hinder their performance on reasoning tasks (Berglund et al., 2023). While these early findings offer insight, much remains to be explored to fully understand how document segmentation and packing influence model performance and how to optimize them effectively.

For instance, while packing is primarily adopted for its compute efficiency, it remains unclear whether this is its only benefit. Could the inclusion of more data per gradient step or the possibility of cross-document attention also contribute to improved model performance? If so, this raises further questions about how best to structure packed sequences: What is the optimal method for selecting which documents to pack together? Should cross-document attention be permitted between any pair of documents, or only between those that are semantically related? Additionally, there is the broader question of whether it is better to segment documents to eliminate padding entirely, or to preserve document boundaries to maintain coherence. Importantly, the answers to these questions may not be universal—they could vary depending on the stage of training or the specific context, such as pre-training, fine-tuning for broad capabilities, or specific downstream tasks.

To explore some of these open questions in a concrete setting, we examine how document packing influences a key downstream capability of LLMs: *latent multi-hop reasoning*—the ability to recall and integrate multiple pieces of information from their parametric knowledge to solve a problem (Wang et al., 2024; Treutlein et al., 2024). Using the HotpotQA (Yang et al., 2018) and 2WikiMultiHopQA (Ho et al., 2020) benchmarks, we evaluate several packing strategies on recent open-weight LLMs (Gemma 2 (Gemma Team, 2024) and Llama 3 (Dubey et al., 2024)), and assess their effects on model recall and reasoning performance. Our key findings are as follows:

- **Result 1.** Packing can improve both the accuracy of recalled information—by helping the model identify what to retrieve, reducing hallucinations in the recalled content—and the accuracy of the final answer compared to no packing.
- **Result 2.** Performance exhibits a sweet spot in the number of documents packed per sequence: both too few and too many documents degrade results.
- **Result 3.** Packing increases the compute required, not only because of cross-document attention but also because more documents must be processed during training to achieve convergence.
- **Result 4.** Cross-document attention and repacking at each epoch are essential for packing to yield improvements over no packing.
- **Result 5.** Scaling batch size is not equivalent to scaling the number of packed documents per sequence when cross-document attention is enabled; the two have distinct effects on performance.

We elaborate on these results in detail throughout the paper, together with additional analyses and ablations that further clarify their implications. These insights contribute to a deeper understanding of document packing in LLM training, particularly in enhancing reasoning capabilities. Our findings not only validate the utility of packing but also provide actionable strategies to optimize its implementation in future models.

## 2 BACKGROUND

### 2.1 LATENT MULTI-HOP REASONING

Multi-hop reasoning (Yang et al., 2018; Welbl et al., 2017; Ho et al., 2020) is a form of open-book question answering (Hill et al., 2015; Kočiský et al., 2017; Pang et al., 2021) that challenges a model to integrate multiple pieces of information to arrive at a correct answer. In this task, the model is presented with a question and must search through a collection of documents—such as news articles, Wikipedia entries, books, or scientific papers—to locate the necessary evidence before responding.

To increase the difficulty, the document set often includes distractors: texts that appear relevant to the question but do not actually contain the required information. This compels the model to perform deeper reasoning and more discerning recall. Importantly, there are generally no constraints on how much searching or reasoning the model can engage in before producing its answer.

A variation of this task, latent multi-hop reasoning (Berglund et al., 2023; Wang et al., 2024; Treutlein et al., 2024), falls under the category of closed-book question answering (Petroni et al., 2019; Roberts et al., 2020; Hendrycks et al., 2020; Mallen et al., 2022). Here, the model does not have access to any external documents at inference time. Instead, it is pre-trained on a corpus of documents, effectively internalizing the knowledge within its parameters. When later given a question, the model must rely solely on its memorized knowledge to reconstruct and reason through an answer.

As with the open-book version, the model is typically free to perform as much internal reasoning and recall as necessary before responding. The document corpus used for this pre-training can vary: for instance, it may consist of broad web data used in training general-purpose models like ChatGPT (OpenAI et al., 2023), Claude (Anthropic, 2024), or Gemini (Gemini Team et al., 2024), which often perform latent multi-hop reasoning during user interactions. Alternatively, the corpus might be more specialized—such as an organization continually fine-tuning an LLM model on proprietary internal documents to support employee needs.

### 2.2 DOCUMENT SEGMENTATION & PACKING

To minimize unnecessary padding in training sequences, documents are packed and truncated as needed to fit within the model's context window (Raffel et al., 2019; Brown et al., 2020; Chowdhery et al., 2022). However, prior research has shown that when this process is carried out randomly, it can negatively impact the model's performance on downstream tasks (Shi et al., 2023; Ding et al., 2024) and increase its susceptibility to hallucinations (Prato et al., 2023). Despite these findings, much remains to be understood about how document segmentation and packing influence the quality and capabilities of LLMs. This gap in knowledge motivates our investigation into their impact on latent multi-hop reasoning.

## 3 METHODOLOGY

In this work, we investigate the role of packing during the continual pre-training of LLMs on a document dataset, and how it influences downstream performance on latent multi-hop reasoning tasks. Our goal is to understand whether packing provides a meaningful advantage over not packing, and if so, to identify the mechanisms behind that benefit.

To explore this, we continually pre-train LLMs using the document corpus from a multi-hop reasoning benchmark. We evaluate several packing strategies during continual pre-training, including a baseline condition with no packing. After continual pre-training, we instruction-tune each model on the benchmark's training set of question-answer (Q/A) pairs and assess their performance on the corresponding evaluation set.

Our methodology also involves varying key parameters of the packing process—for instance, toggling cross-document attention and adjusting the number of documents packed together. These experiments allow us to analyze how different aspects of packing influence the model's reasoning capabilities and to isolate the features that contribute most significantly to performance gains.

The subsections that follow describe this methodology in greater detail.

### 3.1 CONTINUAL PRE-TRAINING

For our study, we focus on two well-established multi-hop reasoning benchmarks: HotpotQA (Yang et al., 2018) and 2WikiMultiHopQA (Ho et al., 2020). Both datasets consist of Wikipedia articles, along with training, validation, and test sets of question-answer (Q/A) pairs. Each question in these benchmarks requires reasoning across multiple articles. Additionally, both datasets include distractor articles—documents that are topically relevant but do not contain the necessary information to answer the question—thereby increasing the task's complexity. Examples of documents and Q/A pairs from each benchmark are provided in Appendix A.

Throughout this and following sections, we use the term document to refer to an article in either HotpotQA or 2WikiMultiHopQA. We treat the two benchmarks separately in our experiments rather than merging them into a single dataset.

The first step in our methodology involves continually pre-training an LLM on the benchmark's document corpus, with the goal of internalizing this information. Since we work with decoder-only LLMs, we use causal language modeling—i.e., next-token prediction—mirroring their original pre-training process. The key difference is that we experiment with various packing methods during this phase.

During pre-training, documents are concatenated sequentially until the model's context window is filled, with the final document truncated if necessary. A special <SEP> token is inserted between documents to denote boundaries. Cross-document attention is disabled, meaning tokens within one document cannot attend to tokens in other documents. However, since our downstream task involves multi-hop reasoning across documents, we enable cross-document attention during continual pre-training—except in ablation studies where its effect is explicitly evaluated (see Section 4.3).

We also explore the impact of varying how many documents are packed into each sequence. Rather than always filling the context window, we examine three distinct strategies:

1. **Fixed-packing:** A set number of documents, $x$, are packed per sequence. For example, *Pack 2* means each training sequence includes two documents.
2. **Multi-packing:** The number of documents per sequence is randomly chosen from a predefined set. For example, *Pack 2-4-8* means each training sequences contains either two, four, or eight documents.
3. **No packing:** Each sequence contains only a single document.

An illustrative comparison of these strategies is shown in Figure 1.

Document selection for packing is guided by the benchmark datasets, which provide, for each question, a list of associated articles—including both relevant and distractor documents. We sample documents at random from each list to construct our sequences. For example, if a list includes ten documents and we use the Pack 2 strategy, we create five sequences, each containing two documents.

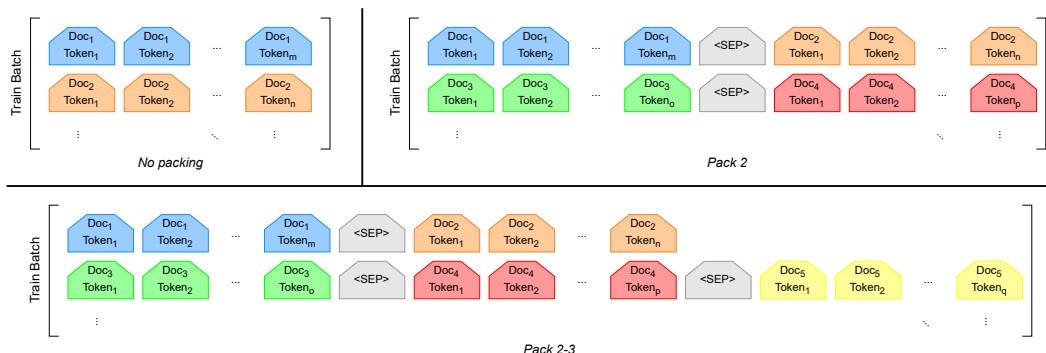

Figure 1: Illustrative example of different packing strategies. (Top-left) *No packing:* Each training sequence contains only one document. (Top-right) *Pack 2:* Two documents are concatenated per sequence, separated by a <SEP> token. (Bottom) *Pack 2-3:* The number of documents per sequence is either two or three, determined at random.

At the beginning of each training epoch, we iterate through all such lists to generate sequences, then pool them into a single set. From this set, we randomly sample sequences without replacement to form training batches. This process is repeated at the start of every epoch to ensure variability in how documents are packed, preventing the model from memorizing specific sequence structures.

Training continues until convergence in training loss, after which we proceed to the instruction-tuning phase.

## 3.2 INSTRUCTION-TUNING

To evaluate the impact of packing strategies during continual pre-training on downstream latent multi-hop reasoning, we assess model performance using the question-answer sets provided by each benchmark. Specifically, we frame this evaluation as an instruction-tuning task (Ouyang et al., 2022; Chung et al., 2022; Zhang et al., 2023): given a question (used as the prompt), the model must first recall relevant documents from its parametric knowledge and then provide an answer to the question.

This setup constitutes a closed-book question answering task. The model cannot access any external documents at inference time; it must rely entirely on information stored within its parameters. The expected output from the model follows a structured format:

```
Recalled Article 1:  ...
Recalled Article 2:  ...
.
.
.
Answer:  ...
```

The model is expected to recall both the title and content of each article. This recall process serves as a form of chain-of-thought reasoning, providing a structured intermediate step that supports the final answer. Our experiments confirm the utility of this format: models that attempt to answer questions without recalling relevant documents consistently perform worse. Examples illustrating this task are provided in the Appendix A.

To train the model for this task, we apply supervised fine-tuning on the benchmark's training set, periodically evaluating on the validation set. During training, the model receives as input a question concatenated with the corresponding ground-truth generation and is trained using causal language modeling to predict the tokens in the generation segment.

At evaluation time, the model is prompted with only the question and must generate the full output, including the recalled documents and final answer. We assess the quality of the model's output using three key metrics:

- **Precision:** Measures the percentage of article titles recalled by the model that are present in the ground truth. The order of recalled titles is not considered. This metric evaluates the model's ability to identify which documents are relevant, regardless of the accuracy of their content.
- **Hallucination Rate:** Among the articles for which the title is correctly recalled (i.e., matches the ground truth), we report the percentage whose content does not match the ground truth. This measures the accuracy of the model's document recall beyond just the title.
- **Accuracy:** Assesses whether the final answer matches the ground-truth answer. Due to the open-ended nature of the responses, we use an LLM as a judge to evaluate semantic correctness. Further details on this procedure are provided in the Appendix C.

Examples of model generations and their corresponding scores can also be found in the Appendix A. Note that the test sets for both HotpotQA and 2WikiMultiHopQA are not publicly available. Consistent with standard practice, we report results on the validation sets instead. Training continues until validation performance converges.

This evaluation framework enables us to systematically measure the effect of different packing strategies during the continual pre-training phase. Specifically, it allows us to analyze how packing influences (1) the model's ability to select relevant documents (as captured by precision), (2) the accuracy of recalled document content (via hallucination rate), and (3) the overall ability to answer questions correctly (reflected in accuracy).

## 4 EXPERIMENTS

### 4.1 SETUP

We conduct experiments using a variety of packing strategies, including no packing, as well as fixed and multi-packing, exploring a broad range of parameters for each. For every strategy—e.g., Pack 2—we evaluate its performance across all models on both the HotpotQA and 2WikiMultiHopQA datasets, adhering to the methodology outlined in Section 3. The full set of hyperparameters used for each experiment is detailed in Appendix B.

**Datasets.** HotpotQA (Yang et al., 2018) is divided into three subsets based on difficulty: easy, medium, and hard. For our experiments, we focus on the easy subset, which we find to be sufficiently challenging for the models under evaluation. Regarding the 2WikiMultiHopQA (Ho et al., 2020) dataset, due to its substantial size—approximately 1.6 million documents—we use a randomly sampled 10% portion for our experiments.

**Models.** We use widely adopted open-weight language models: Llama 3.2 3B, Llama 3.1 8B (Dubey et al., 2024), and Gemma 2 2B (Gemma Team, 2024). We employ their instruction-tuned variants, as these models have already been optimized for general-purpose question answering and thus require minimal additional adaptation for our downstream task. Empirically, we observe that the instruction-tuned versions consistently outperform their base pre-trained counterparts.

### 4.2 RESULTS

The central question of our study is whether packing influences the latent multi-hop reasoning capabilities of LLMs. Since these models must synthesize information from multiple documents to solve complex reasoning tasks, a key consideration is whether presenting these documents in separate training sequences or within the same sequence affects performance.

Across both datasets, we find that models trained without packing consistently underperform compared to those using certain packing strategies, in terms of precision, hallucination rate and accuracy (Table 1). This suggests that carefully applied packing can provide measurable benefits for LLMs in latent multi-hop reasoning tasks.

**Effect on Precision.** Across both datasets, models are tasked with recalling relevant parametric knowledge documents necessary for answering a given question. However, selecting the correct documents is itself a reasoning challenge, as these datasets include distractor documents that can mislead the recall process.

Table 1: Performance of fine-tuned LLMs using various packing strategies on two latent multi-hop reasoning benchmarks. Our results show that packing can enhance performance compared to no packing.

| Model | Packing Strategy | HotpotQA | | | 2WikiMultiHopQA | | |
|---|---|---|---|---|---|---|---|
| | | Precision | Hal. Rate | Accuracy | Precision | Hal. Rate | Accuracy |
| Gemma 2-2B | No packing | 65.63 | 15.52 | 58.64 | 94.23 | 9.15 | 69.84 |
| | Pack 2 | 70.69 | **11.09** | 63.74 | 96.18 | 2.03 | 75.06 |
| | Pack 4 | **71.80** | 13.40 | **64.63** | 96.53 | **1.32** | 76.68 |
| | Pack 6 | 71.36 | 20.65 | 62.85 | **96.83** | 1.78 | **76.86** |
| | Pack 8 | 69.69 | 27.69 | 62.07 | 96.13 | 2.77 | 75.63 |
| | Pack 10 | 68.41 | 47.40 | 57.73 | 96.12 | 4.90 | 76.15 |
| | Pack 2-4 | 69.17 | 15.04 | 63.07 | 96.37 | 2.36 | 74.85 |
| | Pack 2-8 | 69.56 | 27.68 | 63.18 | 96.00 | 3.30 | 75.39 |
| | Pack 4-8 | 70.13 | 25.66 | 62.74 | 96.23 | 2.26 | 75.42 |
| | Pack 2-4-8 | 69.62 | 24.66 | 62.51 | 96.15 | 2.26 | 75.54 |
| Llama 3.2-3B | No packing | 66.17 | 31.24 | 56.73 | 94.37 | 21.83 | 62.32 |
| | Pack 2 | 69.91 | 14.24 | 64.85 | 96.26 | 5.24 | 76.35 |
| | Pack 4 | 70.86 | 21.43 | 66.85 | 96.66 | **4.55** | 78.53 |
| | Pack 6 | **71.86** | 24.85 | 67.52 | **96.86** | 4.62 | **79.23** |
| | Pack 8 | 69.58 | 28.38 | 66.18 | 96.01 | 7.69 | 77.47 |
| | Pack 10 | 68.63 | 44.49 | 62.63 | 96.08 | 8.46 | 77.33 |
| | Pack 2-4 | 71.36 | **18.32** | **68.63** | 96.48 | 5.45 | 76.63 |
| | Pack 2-8 | 69.30 | 27.21 | 66.96 | 95.99 | 8.69 | 75.99 |
| | Pack 4-8 | 71.13 | 26.51 | 65.63 | 96.27 | 6.59 | 77.69 |
| | Pack 2-4-8 | 70.52 | 25.00 | 65.52 | 96.13 | 6.75 | 77.59 |
| Llama 3.1-8B | No packing | 70.68 | 43.88 | 64.29 | 92.29 | 51.31 | 66.52 |
| | Pack 2 | **76.07** | **30.80** | **72.3** | **96.42** | **21.9** | **77.83** |

To assess the impact of packing on a model's ability to correctly identify the relevant documents, we report the precision metric defined in Section 3. Our findings suggest that packing enhances the capability of LLMs to correctly determine which documents should be recalled in multi-hop reasoning scenarios. However, we observe a "sweet spot": precision seems to increase from Pack 2 to Pack 4 and Pack 6, but gradually declines at Pack 8 and Pack 10. This pattern may indicate that a moderate number of documents allows the model to form useful representations for relevance estimation, whereas excessive packing begins to hinder performance. Whether larger packs would cause further degradation remains unclear.

We perform a qualitative analysis on the recalled titles. For HotpotQA, we find that when a title does not match the ground truth, in half of the cases, the model selects a title from another document in the dataset, indicating a failure in document selection. In the other half, the model hallucinates a title that does not exist in the dataset. As for 2WikiMultiHopQA, hallucinations occur more frequently, with one incorrect document title recalled for every four hallucinated titles. Notably, packing does not alter these error distributions; however, it does improve precision by increasing the rate of correctly recalled titles compared to no packing.

**Effect on Hallucination Rate.** Building on our analysis of how packing influences precision, we also investigate whether it affects a model's ability to accurately recall a document's content. To assess this, we measure the hallucination rate as defined in Section 3.

We find that with a small number of packed documents, the hallucination rate is lower than with no packing. However, as the number of packed documents per sequence increases, the hallucination rate rises and, in some cases, even exceeds that of no packing. This suggests that larger packs may impair the model's ability to retain information about individual documents. One possible explanation is that when more documents are packed, the model's internal representations of the text become increasingly entangled across documents. Rather than preserving clear, document-specific memory traces, the model may instead learn blended or context-dependent patterns that make accurate recall more difficult.

**Effect on Accuracy.** We observed that packing can improve both precision and hallucination rates. By retrieving more relevant documents and reducing hallucinations in the recalled content, the model

Table 2: Number of continual pre-training steps (left) and documents (right) required for training loss convergence of the Gemma 2 2B model on HotpotQA.

| Packing Strategy | BS 32 | BS 64 | BS 128 | BS 256 | Packing Strategy | BS 32 | BS 64 | BS 128 | BS 256 |
|---|---|---|---|---|---|---|---|---|---|
| No packing | 48.8k | 24.4k | 13.7k | 6.1k | No packing | 1.6M | 1.6M | 1.8M | 1.6M |
| Pack 2 | 29.8k | 14.9k | 7.4k | 4.1k | Pack 2 | 1.9M | 1.9M | 1.9M | 2.1M |
| Pack 4 | 22.0k | 11k | 6.6k | 3.3k | Pack 4 | 2.8M | 2.8M | 3.4M | 3.4M |
| Pack 8 | 21.4k | 13.2k | 6.6k | | Pack 8 | 5.5M | 6.8M | 6.8M | |

is exposed more consistently to accurate information—an essential factor for producing correct answers. Together, these effects seemingly lead to measurable gains in overall accuracy.

**How many documents should you pack?**    As shown in Table 1, increasing the number of packed documents per sequence has a non-linear effect: performance improves up to a point but then declines, with both precision and hallucination rate worsening at larger pack sizes. This suggests the existence of a sweet spot, though its exact location likely depends on factors such as model scale, task type, and document length. In our experiments, we observe that packing approximately 4 to 6 documents appears to provide a favorable balance for HotpotQA and 2WikiMultiHopQA.

**Does multi-packing help?**    In fixed-packing, the number of documents per sequence remains constant throughout training. Because the downstream task requires the model to recall as many documents as needed to answer a question, we hypothesized that fixed-packing might bias the model toward recalling a fixed number. To address this, we experimented with multi-packing, where the number of documents per sequence is randomized. However, as shown in Table 1, multi-packing offers no measurable benefit, performing comparably to fixed-packing. This suggests that concerns about overfitting to a fixed recall pattern were unfounded.

**Packing & Model Scale.**    One important question when considering packing is whether its benefits persist as model size increases. On both datasets, we observe that the performance improvement from using packing—as compared to not using it—appears consistent across both Llama 3.1 8B and Llama 3.2 3B. This suggests that, at least within the tested range of model sizes, the advantages of packing persist.

**Effect on Rate of Convergence.**    Another key question is whether packing influences the convergence rate of the models. In Table 2, we report both the number of continual pre-training steps (left) and the total number of documents processed (right) required for the Gemma 2 2B model to converge on HotpotQA.

We find that doubling the batch size roughly halves the number of steps to convergence. Doubling the number of packed documents per sequence also reduces the number of steps, but the effect is weaker and exhibits diminishing returns. As a result, the total number of documents the model must process increases as packing is scaled, meaning that packing requires more compute than no packing—even before accounting for the additional cost of cross-document attention. One possible explanation is that in the case of packing, documents appear jointly within the same sequence rather than independently, which creates more complex cross-document representations. These richer interactions may require the model to process more documents overall in order to converge. To verify this hypothesis, we experiment with disabling cross-document attention and find that packing then converges in roughly the same number of steps and total documents as no packing (Appendix D).

## 4.3    WHY PACKING HELPS

So far, we have examined the impact of packing on various aspects of latent multi-hop reasoning. While our findings suggest that packing can be beneficial, the underlying reasons for its advantages over an unpacked approach remain unclear. In the following section, we aim to gain a deeper understanding of how it influences training dynamics in comparison to no packing and how these effects contribute to improved downstream performance.

To explore this further, we pose a series of key questions and conduct the following ablation study.

Table 3: Accuracy of Gemma 2 2B on HotpotQA for various packing strategies and batch sizes, with cross-document attention enabled (left) and disabled (right). Cells are shaded in alternating diagonals to improve readability. For one diagonal, ∕, the total number of documents per batch is constant, e.g., no packing with BS 64 ≡ Pack 2 with BS 32.

| Packing Strategy | BS 32 | BS 64 | BS 128 | BS 256 |
|---|---|---|---|---|
| No packing | 58.64 | 60.07 | 59.18 | 59.07 |
| Pack 2 | 63.74 | 65.74 | 65.74 | 65.41 |
| Pack 4 | 64.63 | 65.07 | 64.96 | 65.07 |
| Pack 8 | 62.07 | 63.07 | 63.29 | |

| Packing Strategy | BS 32 | BS 64 |
|---|---|---|
| No packing | 58.64 | 60.07 |
| Pack 2 | 61.18 | 60.29 |

Table 4: Effect of repacking versus reusing the same packed sequences each epoch. We find that repacking is key for packing to outperform no packing.

| Ablation | HotpotQA | | | 2WikiMultiHopQA | | |
|---|---|---|---|---|---|---|
| | Precision | Hal. Rate | Accuracy | Precision | Hal. Rate | Accuracy |
| No packing | 65.63 | 15.52 | 58.64 | 94.23 | 9.15 | 69.84 |
| Repack every epoch | **70.69** | **11.09** | **63.74** | **96.18** | **2.03** | **75.06** |
| No repack | 65.74 | 32.91 | 56.95 | 93.22 | 12.60 | 69.61 |
| No repack, reshuffle doc order | 69.60 | 14.88 | 61.18 | 95.35 | 3.76 | 73.24 |

**Number of Documents per Batch.** A key effect of packing is that it increases the number of documents processed per batch. One might ask whether this is equivalent to simply scaling the batch size in the absence of packing. In Table 1, the batch size was fixed across all experiments, meaning that the total number of documents per batch varied depending on the packing method. To disentangle these effects, we also evaluate scenarios where the batch size is explicitly controlled.

In Table 3 (left), we report downstream accuracy of Gemma 2 2B on HotpotQA across various batch sizes. Along each diagonal (∕), the number of documents per batch is matched across packing configurations. We consistently observe a performance gap, with packing outperforming no packing within the same diagonal. This indicates that the benefit of packing cannot be explained by scaling alone. Moreover, increasing the number of packed documents per sequence has a qualitatively different effect on performance than increasing the batch size.

**Cross-document Attention.** Our prior experiments were conducted with cross-document attention enabled, allowing tokens from one document to attend to tokens in other documents within a packed sequence. This mechanism creates a distinct training dynamic and may explain the performance differences observed.

To test this, we run experiments with cross-document attention disabled and report the results in Table 3 (right). Under this setting, the benefits of packing disappear, with Pack 2 performing comparably to no packing. These findings suggest that the advantage of packing stems from the model's ability to learn contextualized representations across documents, which could be crucial for improving downstream latent multi-hop reasoning.

**Varying Contexts.** We have established that allowing documents to attend to one another is a key factor in the effectiveness of packing. However, this raises an important follow-up question: does the set of documents packed together need to change every epoch, or is it sufficient for documents to consistently appear in the same context throughout training?

To investigate this, we conduct an experiment in which we pack documents only once at the start of training, rather than repacking them each epoch. In Table 4, we present the results for this setting under *No repacking* and compare it to *Repack every epoch*, where the only difference is that repacking occurs every epoch. Cross-document attention is enabled in both settings.

Our findings reveal that not only does the No repacking condition underperform repacking, but it even lags behind the no-packing baseline in terms of hallucination rate and accuracy. This suggests that repeatedly exposing documents to the same context is more detrimental than seeing them in isolation, possibly hindering the model's ability to recall documents that were not presented within

the same context. Thus, varying which documents are packed together each epoch appears to be a crucial factor in the success of packing.

**Varying Document Order.** In the previous ablation, documents were packed only once at the start of training, meaning their order within each training sequence remained fixed throughout. Under the next-token prediction framework, this setup results in a constant attention pattern: the first document in a sequence can attend only to itself, the second document can attend to both itself and the first document, the third can attend to the first two and itself, and so on.

A natural variation of this approach is to reshuffle the order of documents within each training sequence at the start of every epoch. This allows documents to attend to different documents over epochs, introducing more variability in the contexts they are exposed to.

To test this, we conduct an experiment where we retain the No repacking setup but shuffle document order each epoch. We report the results under *No repacking, reshuffle doc order* in Table 4. While this approach does not perform as well as full repacking (Repack every epoch), it significantly out-performs the fixed-order condition (No repacking). This finding aligns with our intuition: although reshuffling does not provide as much contextual diversity as repacking, it still enables the model to learn a more useful representation of the data compared to a strictly fixed document order.

## 5  DISCUSSION

While this work has addressed several questions regarding packing in the context of continual pre-training and latent multi-hop reasoning, many broader issues surrounding packing remain open for exploration.

One central question is how to optimally pack documents. In many multi-hop reasoning datasets, relevant documents are explicitly listed for each question, providing a natural structure for packing. However, this is not the case in other settings—particularly during pre-training—where no such lists are available. Although there has been some initial progress in determining effective document groupings (Shi et al., 2023), the space remains largely underexplored.

Another promising avenue involves reconsidering what should be packed. Instead of entire documents, it may be more effective to pack only the most relevant excerpts. For example, aggregating key paragraphs from various sources could potentially enhance performance on certain downstream tasks. This raises deeper questions about how information should be curated and presented to the model.

Beyond document selection, cross-document attention introduces further complexity. Should the attention mechanism allow interactions across the entire packed sequence, or should it restrict it to only certain documents or passages? This design choice could affect how the model processes, integrates, and retains information.

Finally, our findings suggest that packing can both enhance precision and reduce hallucination rates. This presents a compelling case for researchers working on tasks that rely heavily on recall and memorization—such as leveraging LLMs as knowledge bases—to explore the potential benefits of packing techniques in their domains.

## 6  CONCLUSION

This study aimed to explore the impact of packing on the latent multi-hop reasoning capabilities of LLMs and to identify the key factors driving this effect. Our findings indicate that packing can enhance performance compared to no packing, improving both recall and answer correctness.

As for the underlying factors contributing to these advantages, two elements stood out: the ability of documents to attend to others within the same training sequence and the practice of repacking at every epoch.

Overall, this study sheds light on an important yet underexplored aspect of LLM training. By deepening our understanding of packing's role in latent multi-hop reasoning, we offer valuable insights for the research community and future model development.

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

## A  TASK EXAMPLES

We present examples of a prompt, the ground truth answer, and Gemma 2 2B's answer, along with their corresponding scores, for both HotpotQA and 2WikiMultiHopQA in Table 6 and Table 7, respectively.

## B  HYPERPARAMETERS

All models are trained with the Adam optimizer (Kingma & Ba, 2014), configured with beta values of 0.9 and 0.999 and an epsilon of 1e-8. No weight decay is applied. The learning rate is warmed up from zero to a tuned base value, which depends on the model, batch size, and number of packed documents per sequence. Specifically:

- Gemma 2 2B: 5e-5 to 7e-5
- Llama 3.2 3B: 1e-4 to 1.4e-4
- Llama 3.1 8B: 5e-6 to 7e-6

Gradient clipping is applied with a norm of 10. Experiments in Table 1 and Table 4 use a batch size of 32. As for the remaining experiments, batch sizes are reported alongside their respective results.

## C  EVALUATION OF ANSWERS

When evaluating answers, given the open-ended format, we use an LLM to automatically judge whether each answer matches the ground truth (Zheng et al., 2023; Gu et al., 2024; Li et al., 2024).

Table 5: Number of continual pre-training steps (left) and documents (right) required for training loss convergence of the Gemma 2 2B model on HotpotQA, when cross-document attention is **disabled**.

| Packing Strategy | BS 32 | BS 64 | Packing Strategy | BS 32 | BS 64 |
|---|---|---|---|---|---|
| No packing | 48.8k | 24.4k | No packing | 1.6M | 1.6M |
| Pack 2 | 26.5k | 13.3k | Pack 2 | 1.7M | 1.7M |

Specifically, for each question, we prompt a Llama 3.1 8B instruction-tuned model[1] with the following instruction:

```
Your task is to compare the model's answer to the
expected answer and determine if the model's answer
is correct.  Respond with ``yes'' if the answer
is correct, and ``no'' if it is incorrect.  Do not
include any explanations.

Question:  ...
Expected Answer:  ...
Model's Answer:  ...
```

We record the number of "yes" responses out of the total number of questions. In practice, we observe that the model consistently responds with either "yes" or "no", as instructed. To ensure the reliability of this approach, we manually review a subset of the answers and find the prediction to be accurate, especially given the straightforward nature of the answers (e.g., dates, names).

## D EFFECT OF CROSS-DOCUMENT ATTENTION ON THE CONVERGENCE RATE

In Section 4.2, we observed that packing required processing a greater number of documents during training for the model to converge. We hypothesize that this is due to cross-document attention, which provides richer contexts for the model to learn. To test this hypothesis, we disable cross-document attention and examine its effect on convergence. The results are reported in Table 5, with the total number of continual pre-training steps shown on the right and the total number of documents processed shown on the left. We find that with cross-document attention disabled, both Pack 2 and no packing converge in approximately the same number of steps when the number of documents per batch is matched ($\diagup$). Likewise, the total number of documents processed to reach convergence is also comparable across conditions. These results support our hypothesis that cross-document attention accounts for the differences observed in convergence behavior.

## E LLM USAGE

ChatGPT 4o and 5 were used to enhance the writing quality of this paper.

---

[1]https://huggingface.co/meta-llama/Llama-3.1-8B-Instruct

Table 6: Example from the HotpotQA benchmark showing the prompt, ground truth, and model-generated answer. The first article includes a hallucinated fact and an incorrect punctuation mark. In the second article, the model hallucinates a paragraph about the Gotham TV series instead of referencing the correct article on season 4 of the show. Despite these issues, the final answer is correct. This yields a precision of 50%, a hallucination rate of 100%, and an accuracy of 100%.

---

**Prompt Example**

---

Below is a question. Your task is to read the question, recall the necessary information, and provide a concise answer. Please ensure your answer is based only on the recalled information.

# Question:
"A Dark Knight: The Fear Reaper" is the second episode of the fourth season, of the American television series "Gotham", based on characters, from which company?

---

**Ground Truth**

---

# Evidence:
## A Dark Knight: The Fear Reaper
"A Dark Knight: The Fear Reaper" is the second episode of the fourth season and 68th episode overall from the Fox series "Gotham". The show is itself based on the characters created by DC Comics set in the Batman mythology. The episode was written by executive producer Danny Cannon and directed by Louis Shaw Milito. It was first broadcast on September 28, 2017.

## Gotham (season 4)
The fourth season of the American television series "Gotham", based on characters from DC Comics related to the Batman franchise, revolves around the characters of James Gordon and Bruce Wayne. The season is produced by Primrose Hill Productions, DC Entertainment, and Warner Bros. Television, with Bruno Heller, Danny Cannon, and John Stephens serving as executive producers. The first half of the season will be inspired by the comic book story ", and the second half by ". The subtitle for the first half of the season is "A Dark Knight".

# Answer:
DC Comics

---

**Model Answer**

---

# Evidence:
## A Dark Knight: The Fear Reaper
"A Dark Knight: The Fear Reaper" is the second episode of the fourth season, and 67th episode overall from the Fox series "Gotham". The show is itself based on the characters created by DC Comics set in the Batman mythology. The episode was written by executive producer Danny Cannon and directed by Louis Shaw Milito. It was first broadcast on September 28, 2017.

## Gotham (TV series)
Gotham is an American crime drama television series developed by Bruno Heller, based on characters published by DC Comics and appearing in the Batman franchise, primarily those of James Gordon and Bruce Wayne. The series stars Ben McKenzie as the young Gordon, while Heller executive-produces, along with Danny Cannon, who also directed the pilot.

# Answer:
DC Comics

---

Table 7: Example from the 2WikiMultiHopQA benchmark showing the prompt, ground truth, and model-generated answer. The model correctly recalls the titles of all four articles and the content of three. Its final answer is also correct. This results in a precision of 100%, a hallucination rate of 75%, and an accuracy of 100%.

---

**Prompt Example**

---

Below is a question. Your task is to read the question, recall the necessary information, and provide a concise answer. Please ensure your answer is based only on the recalled information.

# Question:
Which film whose director is younger, Dallas 362 or Revenge Of The Barbarians?

---

**Ground Truth**

---

# Evidence:
## Dallas 362
Dallas 362 is a 2003 film, starring and directed by Scott Caan. This film was Caan's debut as a director. The movie won the Critics Award at the 2003 CineVegas International Film Festival in Las Vegas, Nevada.

## Revenge of the Barbarians
Revenge of the Barbarians is a 1960 film about the sack of Rome in AD 410 by the Visigoths. This film was written by Gastone Ramazzotti and directed by Giuseppe Vari.

## Scott Caan
Scott Andrew Caan (born August 23, 1976) is an American actor. He stars as Detective Sergeant Danny "Danno" Williams in the CBS television series "Hawaii Five-0" (2010 – present), for which he was nominated for a Golden Globe Award. Caan had a recurring role as manager Scott Lavin in the HBO television series "Entourage" (2009 – 2011). In the 1990s, he was a part of hip hop group The Whooliganz with The Alchemist. The duo went by the names Mad Skillz and Mudfoot, respectively.

## Giuseppe Vari
Giuseppe Vari (9 March 1924 – 1 October 1993) was an Italian film director, editor and screenwriter.

# Answer:
Dallas 362

---

**Model Answer**

---

# Evidence:
## Dallas 362
Dallas 362 is a 2003 film, starring and directed by Scott Caan. This film was Caan's debut as a director. The movie won the Critics Award at the 2003 CineVegas International Film Festival in Las Vegas, Nevada.

## Revenge of the Barbarians
Revenge of the Barbarians is a 1960 film about the sack of Rome in AD 410 by the Visigoths. This film was written by Gastone Ramazzotti and directed by Giuseppe Vari.

## Scott Caan
Scott Andrew Caan (born August 23, 1966) is an American actor, producer, director, and screenwriter. He first achieved notability for starring in the 1988 zombie horror film "Return of the Living Dead Part II", and for his role as Sean Mitchell in the 1990s ABC/ CBS series "The New Adventures of Little House on the Prairie". He is also known for his roles in the films "Cobra Kai" (2018–2019) and "The Karate Kid" (1995).

## Giuseppe Vari
Giuseppe Vari (9 March 1924 – 1 October 1993) was an Italian film director, editor and screenwriter.

# Answer:
Dallas 362

---

