# OpenReview forum: "Effect of Document Packing on the Latent Multi-Hop Reasoning Capabilities of Large Language Models"
_ICLR.cc/2026/Conference — Submitted to ICLR 2026_

### Official Review · Reviewer_tn2x · 2025-10-25

**Soundness:** 3
**Presentation:** 2
**Contribution:** 2
**Rating:** 4
**Confidence:** 3

**Summary:**

The paper investigates whether document packing during continual pre-training improves latent multi-hop reasoning. Using HotpotQA and 2WikiMultiHopQA with Gemma-2-2B, Llama-3.2-3B, and Llama-3.1-8B, the authors vary the number of packed documents, cross-document attention, repacking frequency, and batch size. They find that packing 4-6 documents improves recall precision, reduces hallucination (conditional on correct titles), and raises answer accuracy, especially when cross-document attention is enabled and repacking occurs each epoch. However, packing increases total compute, and gains diminish without cross-document attention.

**Strengths:**

- Clear empirical question with systematic ablations (packing vs. batch vs. repacking).
- Practical insight that repacking each epoch is crucial for performance.
- Honest discussion of compute–quality trade-offs.
- Consistent “sweet spot” trend (4–6 documents) across settings.

**Weaknesses:**

- The attention masking is ambiguous: §3.1 contains contradictory statements about whether cross-document attention is enabled, leaving the default mask unclear.

- The evaluation scope is conveniently narrow: experiments center on HotpotQA-easy and a 10% 2Wiki subsample with oracle per-question packing, which limits external validity.

- The metrics are biased: hallucination is computed only conditional on correct title selection, and an LLM-as-judge substitutes for standard EM/F1, skewing the interpretation of gains.

- Compute accounting is incomplete: comparisons are not matched on tokens or FLOPs and omit wall-clock or energy, so improvements may largely reflect extra compute rather than packing.

Statistical rigor and reproducibility are weak: there are no multiple seeds, error bars, or significance tests, and key artifacts (packing/masking scripts and exact hyperparameters) are not released.

**Questions:**

- What exact attention mask is used during continual pre-training, are tokens allowed to attend across <SEP> boundaries, and is any block-sparse structure applied beyond causal masking?

- Can you reproduce the main gains under compute-matched conditions (equal tokens/FLOPs), and report wall-clock time and energy to rule out extra compute as the driver?

- Do the improvements persist with non-oracle packing (random/semantic global packing) and on harder multi-hop benchmarks (e.g., HotpotQA medium/hard, MuSiQue, StrategyQA)?

---

> ### Author Response · Authors · 2025-11-28
> **Author Response (1/2)**
>
> We thank the reviewer for the thoughtful feedback.
>
> ---
>
> **Question:** “What exact attention mask is used during continual pre-training”
>
> **Answer:** We use the standard causal language-modeling attention-mask, and enable cross-document attention for all of our experiments, except for the ablation comparing enabling versus disabling cross-document attention from line 411 to 419. Specifically, when cross-document attention is enabled, a given token can attend to prior tokens in the same document as well as tokens from prior documents in the sequence. No token can attend to latter tokens, as per the causal language modeling setup. When disabling cross-document attention, a token can solely attend to prior tokens from the same document. In both cases, cross-document attention enabled or disabled, no additional block-sparse or structured attention is applied beyond the causal mask; the only modification is whether tokens can attend across document boundaries. We hope that this clarifies the setup. We recognize the ambiguity arises because §3.1 first describes pretraining without cross-document attention before specifying that it is enabled for all main runs. We will clarify this ordering.
>
> ---
>
> **Question:** “Can you reproduce the main gains under compute-matched conditions (equal tokens/FLOPs), and report wall-clock time and energy to rule out extra compute as the driver?”
>
> **Answer:** We thank the reviewer for this question. As stated in the paper (line 185), all models are trained until convergence in training loss. We view convergence as the most meaningful comparison criterion, since stopping earlier to match FLOPs would produce under-trained models and obscure genuine performance differences.
>
> We acknowledge that packing with cross-document attention increases per-step FLOPs, but since both setups are trained to convergence, the comparison reflects the quality of the resulting solutions rather than raw compute expenditure.
>
> As an alternative, reviewer DGex suggested including accuracy-per-FLOP for a normalized comparison. We will include both raw and accuracy-per-FLOP in Table 2, as well as clarify this reasoning in the revision.
>
> ---
>
> **Question:** “Do the improvements persist with non-oracle packing”
>
> **Answer:** We ran a Gemma-2-2B model on HotpotQA with Pack 2 (batch size 32) using random global packing, where documents were sampled uniformly from the entire HotpotQA corpus rather than from per-question lists. The table below compares this setting to No Packing and Pack 2 (oracle) packing.
> |  | Precision | Hal. Rate | Accuracy |
> | ----- | ----- | ----- | ----- |
> | No packing | 65.63 | 15.52 | 58.64 |
> | Pack 2, oracle packing | 70.69 | 11.09 | 63.74 |
> | Pack 2, random global packing | 68.24 | 13.20 | 62.18 |
> Performance under random global packing remains higher than No Packing but slightly below oracle packing. This indicates that the improvement is not solely due to oracle grouping: even without it, packing and cross-document attention contribute meaningful gains.
>
> We will include these results and a discussion of their implications in the revision.
>
> ---
>
> **Comment:** “The evaluation scope is conveniently narrow: experiments center on HotpotQA-easy and a 10% 2Wiki subsample with oracle per-question packing, which limits external validity.”
>
> **Response:** We thank the reviewer for noting the scope of our evaluation. Our primary goal was to conduct the first systematic investigation of how document packing affects latent multi-hop reasoning, rather than to achieve exhaustive coverage across all benchmarks. We intentionally selected HotpotQA and 2WikiMultiHopQA because they are well-established, publicly available, and explicitly annotated for multi-hop reasoning chains, which makes them ideal for controlled ablations across multiple variables (packing size, cross-document attention, repacking frequency, batch size).
>
> We agree that extending to more challenging or differently structured datasets (e.g., HotpotQA-medium/hard, MuSiQue, StrategyQA) would further validate generality. However, given the large experimental grid already explored (3 model scales × multiple packing and attention settings × repacking and batch ablations), a full replication on additional datasets would be computationally prohibitive.
>
> That said, we view this work as an initial, foundational study establishing a methodological and empirical baseline. We will make this positioning explicit in the revision and encourage future work to extend these analyses to broader benchmarks. If space allows, we can also include preliminary runs on one additional dataset (e.g., HotpotQA-medium) to demonstrate consistency of trends.

---

> ### Author Response · Authors · 2025-11-28
> **Author Response (2/2)**
>
> **Comment:** “The metrics are biased: hallucination is computed only conditional on correct title selection, and an LLM-as-judge substitutes for standard EM/F1, skewing the interpretation of gains.”
>
> **Response:** We appreciate the reviewer’s concern regarding evaluation metrics. We respectfully disagree that our approach biases interpretation. Our goal was to measure specific, interpretable aspects of latent multi-hop reasoning, (1) document selection, (2) factual recall within those documents, and (3) final answer correctness. Computing hallucination conditional on correct title recall isolates the model’s factual consistency from its retrieval accuracy, allowing each dimension to be assessed independently. This decomposition makes it possible to disentangle what part of reasoning improves with packing (retrieval vs. factual fidelity).
>
> Regarding the use of an LLM-as-judge for answer evaluation, we follow established practice in recent literature (e.g., https://arxiv.org/abs/2306.05685; https://arxiv.org/abs/2411.15594; https://arxiv.org/abs/2412.05579; Appendix C) The model is used only to assess semantic correctness, and we manually verified a subset of judgments. We will clarify this rationale and provide examples in the revision.
>
> We emphasize that because our metric definitions are explicit, comparisons and conclusions are internally consistent and do not rely on hidden assumptions.
>
> ---
>
> **Comment:** “comparisons are not matched on tokens”
>
> **Response:** As shown in Table 2, the Gemma 2 2B model with No Packing and Pack 2 converged after approximately the same number of documents (1.8 M vs 1.9 M, batch size 128). Despite this near-matched token exposure, Pack 2 achieves substantially higher accuracy (65.74 vs 59.18, Table 3) when cross-document attention is enabled.
>
> When cross-document attention is disabled, convergence and performance become comparable (Table 3 and Appendix D), indicating that the improvements arise from cross-document context rather than increased token count.
>
> ---
>
> **Comment:** “there are no multiple seeds”
>
> **Response:** We thank the reviewer for noting this. We did run multiple seeds for several experiments. Specifically, for 2WikiMultiHopQA, we averaged results over five random seeds for the Gemma-2-2B model under No Packing, Pack 2, Pack 6, Pack 10, and Pack 2–10, and the values reported in Table 1 reflect these averages.
>
> For other configurations, we conducted single-seed runs due to the large experimental grid (models × packing × batch × attention). In the multi-seed runs, the standard error of the mean (SEM) was consistently below 0.4, indicating minimal variance across seeds. The relative ranking of packing strategies remained stable across all runs. Specifically, the sweet spot at 4–6 packed documents was consistent across seeds, confirming the robustness of our core finding. We will clarify this in the revision and include these variance estimates where space permits, e.g., we will include mean ± SEM bars for Table 1 in the appendix.
>
> ---
>
> **Comment:** “key artifacts (packing/masking scripts and exact hyperparameters) are not released.”
>
> **Response:** We thank the reviewer for pointing this out. We will release all code and artifacts, including the document packing and masking scripts, upon acceptance of the paper to ensure full reproducibility.
>
> Regarding hyperparameters, all training and optimization settings are already listed in Appendix B. If the reviewer was referring to additional parameters not covered there (e.g., data preprocessing or environment details), we would be happy to provide them in the revision.
>
> ---
>
> We once again thank the reviewer for their thoughtful feedback and constructive comments. We hope that our responses and clarifications have addressed the concerns raised. Should any questions remain, we would be happy to provide further clarification. We also hope that our explanations will be reflected in the reviewer’s final assessment. Thank you for your consideration.

---

### Official Review · Reviewer_DGex · 2025-10-31

**Soundness:** 3
**Presentation:** 3
**Contribution:** 3
**Rating:** 6
**Confidence:** 3

**Summary:**

This paper investigates the impact of document packing on models’ latent multi-hop reasoning capabilities. While document packing is widely used to reduce padding and improve throughput, its effect on learning dynamics has been underexplored. The authors conduct systematic experiments by continually pre-training Llama 3 and Gemma 2 models on HotpotQA and 2WikiMultiHopQA corpora under different packing strategies (fixed and multi-packing) and ablations (cross-document attention, repacking frequency, batch size scaling). They then instruction-tune models on QA tasks that probe latent multi-hop reasoning. The study finds that (1) packing can improve reasoning accuracy and reduce hallucinations; (2) there exists an optimal range (4–6 documents per sequence); (3) packing requires more compute; and (4) cross-document attention and repacking each epoch are critical for gains. The authors conclude that packing may serve not only efficiency purposes but also enhance reasoning through richer inter-document contexts.

**Strengths:**

The paper addresses a genuinely underexplored but important topic—how document packing affects the quality (not just efficiency) of LLM training. Given the ubiquity of packing in large-scale pre-training, this study is timely and of high practical relevance.

The authors explore multiple factors (packing granularity, cross-document attention, repacking vs. fixed contexts, batch size effects) and cleanly isolate their roles. The diagonal comparison in Table 3 effectively demonstrates that packing is not equivalent to batch-size scaling.

The two-stage evaluation is conceptually well-aligned with the multi-hop reasoning objective, and metrics such as precision, hallucination rate, and accuracy are sensibly defined.

The discovery that cross-document attention and repacking frequency are key drivers of performance, and that excessive packing harms reasoning, provides actionable insights for practitioners training large models.

The paper is well-written, carefully structured, and empirically grounded, which facilitates comprehension of a complex systems–reasoning intersection.

**Weaknesses:**

what is the evidence showing that cross-document attention yields richer contextual representations (e.g., attention map analysis, representational similarity, or probing). Without such analysis, the causal link between packing and improved reasoning remains speculative.

Table 2 shows that packing increases compute, but there is no normalized comparison (e.g., accuracy per FLOP). Since the main selling point of packing is efficiency, the practical utility of adopting more expensive packing strategies remains uncertain.

**Questions:**

It is not clearly stated how does the training data look like, whether training data overlap with evaluation data (HotpotQA and 2WikiMultiHopQA corpora).

why with cross-document attention enabled packing ends up increasing total compute to reach convergence.

Why does repacking outperform reshuffling?

Could the authors analyze attention patterns or activation similarities across packed documents to directly support the claim that cross-document attention enhances latent reasoning?

---

> ### Author Response · Authors · 2025-11-28
> **Author Response (1/2)**
>
> We thank the reviewer for the thoughtful feedback.
>
> ---
>
> **Comment:** “what is the evidence showing that cross-document attention yields richer contextual representations (e.g., attention map analysis, representational similarity, or probing). Without such analysis, the causal link between packing and improved reasoning remains speculative.”
>
> **Response:** We appreciate this point and agree that a more direct mechanistic analysis (e.g., probing or attention visualization) would provide valuable evidence. While our current work focuses primarily on empirical performance outcomes, we do provide indirect indications that cross-document attention improves contextual integration. Specifically, models trained with cross-document attention show consistent improvements in precision and reduced hallucination rates (Table 3), which we interpret as evidence of more coherent multi-document reasoning.
>
> We have taken care to present this interpretation as hypothetical rather than causal. For example, in Lines 299, 317, 364, 417, and 446, our phrasing explicitly reflects this speculative stance. We view our findings as suggestive evidence that packing facilitates inter-document contextualization, and we plan to explore this mechanism in future work (e.g., via representational probing and attention entropy analysis).
>
> If the reviewer believes this point should be further clarified in the revision, we are happy to make the speculative nature of our interpretation more explicit in the final version.
>
> ---
>
> **Comment:** “Table 2 shows that packing increases compute, but there is no normalized comparison (e.g., accuracy per FLOP). Since the main selling point of packing is efficiency, the practical utility of adopting more expensive packing strategies remains uncertain.”
>
> **Response:** Thank you for this helpful suggestion. Reviewer tn2x also requested FLOP counts in Table 2, and we will include both raw and accuracy-per-FLOP normalized results in the revised version, for both cross-document attention enabled and disabled. We would also like to clarify that packing serves two different purposes depending on whether cross-document attention is enabled.
> - With cross-document attention disabled, the primary benefit of packing is efficiency (reduced padding and higher throughput).
> - With cross-document attention enabled, packing instead improves downstream reasoning performance, but this comes at additional compute cost.
>
> We will make this distinction clearer in the revision.
>
> ---
>
> **Question:** “It is not clearly stated how does the training data look like, whether training data overlap with evaluation data (HotpotQA and 2WikiMultiHopQA corpora).”
>
> **Answer:** Thank you for raising this point. For continual pre-training, we use only the documents (no Q/A pairs) from the HotpotQA-easy subset and from a 10% sample of 2WikiMultiHopQA, as described in the paper. These datasets are not merged: all experiments on HotpotQA and 2Wiki are conducted separately.
>
> During instruction tuning, we follow the standard dataset splits, training only on the training Q/A pairs and evaluating on the validation Q/A pairs. Furthermore, no document appears in both the training and validation Q/A sets within each dataset. We will make these details explicit in the revised paper.
>
> ---
>
> **Question:** “why with cross-document attention enabled packing ends up increasing total compute to reach convergence.”
>
> **Answer:** We appreciate this question. Enabling cross-document attention increases the per-step compute cost because the attention mechanism must consider a larger effective context (i.e., interactions across all documents in the packed sequence). Beyond this, our results show that the optimization dynamics also change: when cross-document attention is enabled, models benefit from training on more packed documents before converging. Our interpretation, stated explicitly as speculation in Lines 364–369, is that cross-document attention encourages the model to form more complex inter-document representations, which may require additional updates to stabilize during pre-training. However, we do not make any causal claims, as determining the precise mechanism would require targeted analyses (e.g., representational probing or attention-pattern studies), which we leave for future work.

---

> ### Author Response · Authors · 2025-11-28
> **Author Response (2/2)**
>
> **Question:** “Why does repacking outperform reshuffling?”
>
> **Answer:** Thank you for the question. The key difference is that repacking creates new document co-occurrence groups, whereas reshuffling only changes the order of documents within the same fixed groups. For example, if a packed sequence contains documents x_i and  x_j in epoch 1, repacking in the next epoch may pair x_i with x_k or x_m, exposing the model to new cross-document combinations. Reshuffling, in contrast, will always keep x_i and x_j together, only permuting their order. Over many epochs, repacking therefore provides greater contextual diversity, which we hypothesize enables the model to learn more robust cross-document representations. While this remains an intuitive explanation, our empirical results consistently show that repacking leads to better downstream performance (Table 4).
>
> ---
>
> **Question:** “Could the authors analyze attention patterns or activation similarities across packed documents to directly support the claim that cross-document attention enhances latent reasoning?”
>
> **Answer:** We appreciate this suggestion and agree that a mechanistic analysis of attention patterns or activation structure would provide valuable insight. However, this type of analysis is substantial in scope and, given the number of experiments already included, falls beyond what we can reasonably incorporate into the present work. Our claims regarding cross-document contextualization are therefore framed as hypotheses rather than causal conclusions. We view mechanistic investigation as an important direction for future work, and we will explicitly note this in the revised version.
>
> ---
>
> We once again thank the reviewer for their thoughtful feedback and constructive comments. We hope that our responses and clarifications have addressed the concerns raised. Should any questions remain, we would be happy to provide further clarification. We also hope that our explanations will be reflected in the reviewer’s final assessment. Thank you for your consideration.

---

### Official Review · Reviewer_wT9R · 2025-11-01

**Soundness:** 2
**Presentation:** 3
**Contribution:** 2
**Rating:** 4
**Confidence:** 4

**Summary:**

This paper explores whether packing documents together in a training sequence and enabling cross-document attention can improve the latent multi-hop performance of models. To achieve this, the authors continually pre-train three different models, using the documents from multi-hop datasets. They then instruction fine-tune those models to be able to perform multi-hop question-answering. They show that using packing (up to a certain amount of documents packed together) and enabling cross-document attention leads to substantial gains in latent multi-hop performance.

**Strengths:**

- The paper is well-written, easy to follow, and well-motivated. The importance of knowing how to pack documents to achieve the best knowledge retrieval from multiple documents at the same time is an important problem to solve.
- The experiments are thorough, and multiple ablations were carried out to justify the different choices (such as re-packing, or cross-attention). The paper shows that using packing during leads to substantial increases in performance in multi-hop question answering.

**Weaknesses:**

The main weakness of this paper is that the proposed continual pre-training seems to be closer to fine-tuning than actual pre-training. By packing documents that are assigned to a specific question in the dataset, and then performing cross-attention between them, the model is learning that if it retrieves one of the correct documents, then the model knows from learning to output the next document, which one is relevant. This could be seen as a form of leaking test information to the dataset. The results from the paper could be seen as reinforcing this point since packing a lot of documents (Pack 10) would make it harder for the model to know which k documents to select and increase the possible permutations. Also, it would explain the increase in hallucination seen in the no repacking since the model would expect to output a certain value article but gets a different title.

**Questions:**

- How many training epochs of the HotPotQA dataset does each packing strategy represent? (aka how many times is the dataset repeated)
- Were any other methods for choosing which documents to pack together tested?
- For the instruction-tuning and results, was there a split of the validation set held out to test on and get final scores from?
- For no repacking, was a larger packing tested (Pack 8 for example) to see if the increase in hallucination still exists?
- Was the 8B model trained for fewer steps than the others, or the same amount?

---

> ### Author Response · Authors · 2025-11-24
> **Authors' Response**
>
> We thank the reviewer for the thoughtful feedback. We would like to clarify a misunderstanding about our continual pre-training setup. During this phase, the model performs unsupervised next-token prediction on mixed document sequences that include both gold and distractor documents for each question. Documents are randomly sampled and repacked every epoch, so gold documents may or may not co-occur, and the model has no supervision signal indicating which are relevant. Therefore, the continual pre-training phase cannot leak question–answer information or bias the model toward specific document combinations. This is described in Section 3.1 of the paper (lines 159–184).
> > Document selection for packing is guided by the benchmark datasets, which provide, for each question, a list of associated articles---including both relevant and distractor documents. We sample documents at random from each list to construct our sequences. For example, if a list includes ten documents and we use the Pack 2 strategy, we create five sequences, each containing two documents. At the beginning of each training epoch, we iterate through all such lists to generate sequences, then pool them into a single set. From this set, we randomly sample sequences without replacement to form training batches. This process is repeated at the start of every epoch to ensure variability in how documents are packed, preventing the model from memorizing specific sequence structures.
>
> If this distinction between continual pre-training and fine-tuning was not sufficiently clear, we are happy to revise the paper to make it more explicit.
>
> ---
>
> **Question:** “How many training epochs of the HotPotQA dataset does each packing strategy represent?”
>
> **Answer:** The following table shows the number of training epochs of Gemma 2 2B on HotpotQA:
> | Packing Strategy | BS 32 | BS 64 | BS 128 | BS 256 |
> | ----- | ----- | ----- | ----- | ----- |
> | No packing | 9.0 | 9.0 | 10.1 | 9.0 |
> | Pack 2 | 10.7 | 10.7 | 10.7 | 11.8 |
> | Pack 4 | 15.7 | 15.7 | 19.1 | 19.1 |
> | Pack 8 | 30.9 | 38.2 | 38.2 | |
> We will add this information to Table 2 in the final version for completeness.
>
> ---
>
> **Question:** “Were any other methods for choosing which documents to pack together tested?”
>
> **Answer:** No. For this study, we fixed the document-selection method across all experiments to isolate the effects of packing itself. Documents were sampled at random from each question’s pool of gold + distractor articles (as described in Section 3.1). Our goal was to study how packing parameters, such as the number of documents per sequence, enabling or disabling cross-document attention, and whether sequences were repacked each epoch, affect performance. Exploring alternative document-selection heuristics is an interesting direction that we plan to investigate in future work.
>
> ---
>
> **Question:** “For the instruction-tuning and results, was there a split of the validation set held out to test on and get final scores from?”
>
> **Answer:** No. Consistent with standard practice for HotpotQA and 2WikiMultiHopQA, we report results on the validation set, as the official test sets are not publicly available (see line 227). We observed that validation performance was stable and exhibited smooth convergence across runs, suggesting that an additional held-out subset would yield similar results. We will clarify this point in the revised manuscript to avoid confusion.
>
> ---
>
> **Question:** “For no repacking, was a larger packing tested (Pack 8 for example) to see if the increase in hallucination still exists?”
>
> **Answer:** We ran the requested experiment (No repacking, Pack 8), and obtained the following result:
> | Ablation | Packing Strategy | Precision | Hal. Rate | Accuracy |
> | ----- | ----- | ----- | ----- | ----- |
> | No packing | None | 65.63 | 15.52 | 58.64 |
> | Repack each epoch | Pack 2 | 70.69 | 11.09 | 63.74 |
> | No repack | Pack 2 | 65.74 | 32.91 | 56.95 |
> | No repack | Pack 8 | 61.57 | 62.87 | 49.50 |
> | No repack, reshuffle doc order | Pack 2 | 69.6 | 14.88 | 61.18 |
> As observed, precision degraded and the hallucination rate increased substantially, resulting in a degradation in accuracy. These results will be added to the paper.
>
> ---
>
> **Question:** “Was the 8B model trained for fewer steps than the others, or the same amount?”
>
> **Answer:** The 8B model is trained for roughly 50% more training steps to ensure that the model converges.
>
> ---
>
> We once again thank the reviewer for their thoughtful feedback and constructive comments. We hope that our responses and clarifications have addressed the concerns raised. Should any questions remain, we would be happy to provide further clarification. We also hope that our explanations will be reflected in the reviewer’s final assessment. Thank you for your consideration.

---

### Author Response · Authors · 2025-12-03
**Summary for AC**

We would like to briefly summarize how the reviews and discussion resolved the main concerns about our submission.

&nbsp;

### Contribution and significance.
The paper provides, to our knowledge, the first systematic empirical study of how document packing during continual pre-training affects latent multi-hop reasoning in LLMs, across multiple open-weight models (Gemma 2 2B, Llama 3.2 3B, Llama 3.1 8B), packing regimes (fixed vs multi-packing, different pack sizes), and ablations (cross-document attention, repacking vs fixed contexts, batch size). We show that packing with cross-document attention and repacking each epoch consistently improves precision, reduces hallucination, and raises downstream answer accuracy, with a clear “sweet spot” at 4–6 documents per sequence.

&nbsp;

### Clarifying the pre-training setup (no information leakage).
Reviewer wT9R expressed concern that our continual pre-training setup might be closer to fine-tuning, potentially leaking question–answer information via oracle packing. We clarified that continual pre-training is unsupervised next-token prediction over sequences of documents that include both gold and distractor articles. Documents are randomly sampled and repacked every epoch from each question’s pool, and the model never sees questions or answers during this phase, nor any signal about which documents are relevant. As a result, there is no leakage of Q/A information or supervision on particular document combinations; the model simply learns to model text over heterogeneous multi-document contexts. We will make this distinction much more explicit in the revised paper.

&nbsp;

### Addressing concerns about generality of the packing effect.
Reviewer tn2x questioned whether improvements might rely on oracle per-question packing. In response, we ran an additional experiment with *random global packing* (documents sampled uniformly from the entire corpus, rather than per-question pools) on Gemma 2 2B / HotpotQA. We observe that this setting still clearly outperforms no packing, though it is slightly below oracle packing. This indicates that the benefits of packing with cross-document attention are not solely due to oracle grouping, and persist under a more realistic global-packing regime. We will add these results and discussion to the revision.

&nbsp;

### Compute and cross-document attention.
Both DGex and tn2x raised questions about compute normalization and the role of cross-document attention. We (a) clearly distinguish two regimes: packing *without* cross-document attention (primarily an efficiency technique) and *with* cross-document attention (primarily a quality technique), and (b) provide convergence statistics (steps and number of documents) and, in the revision, accuracy-per-FLOP. Our ablations where cross-document attention is disabled show that, when documents per batch are matched, packing then converges in roughly the same number of steps and processed documents as no packing and does not yield accuracy gains. This supports our interpretation that the performance improvements in the main setting are specifically tied to cross-document contextualization rather than simply extra compute.

&nbsp;

### Limitations and positioning.
We fully acknowledge the limitations highlighted by DGex and tn2x: we do not yet provide mechanistic analyses of attention patterns or representations, and the benchmark scope (HotpotQA-easy and a 10% 2Wiki sample) is limited. In the paper and rebuttal, we purposefully frame our explanations about why packing helps (e.g., through cross-document contextualization) as hypotheses rather than strong causal claims, and we clearly identify mechanistic analysis and broader benchmarks as future work. Our goal is to establish a clean empirical baseline and set of design insights for an important training knob, not to claim a fully complete theory.

&nbsp;

Overall, we hope this clarifies that (1) the method is sound and does not rely on leaking Q/A information, (2) the core empirical findings about packing, cross-document attention, and repacking are robust and not limited to an oracle setting, and (3) the remaining concerns are primarily about scope and depth rather than correctness. We believe this makes the paper a reasonable candidate for acceptance as a first, systematic study of how document packing impacts latent multi-hop reasoning in LLMs.

---

### Meta-Review · Area_Chair_7UxJ · 2026-01-02

**Summary:**

This paper presents a systematic empirical study of how document packing during continual pre-training affects the latent multi-hop reasoning capabilities of large language models. Reviewers raised several substantive concerns regarding the interpretation, scope, and evaluation of the proposed study on document packing. The main issues included whether the continual pre-training setup might implicitly leak question-answer information or resemble fine-tuning rather than unsupervised pre-training; whether the observed gains relied on oracle per-question packing and thus lacked generality; and whether performance improvements could be attributed primarily to increased compute rather than to packing or cross-document attention itself. Reviewers also questioned the clarity of the attention masking scheme, the narrow benchmark scope, the lack of mechanistic analysis supporting the role of cross-document attention, and the completeness of compute normalization, statistical rigor, and reproducibility details.

**Reviewer Concerns:**

Reviewer wT9R: Concerns about information leakage and the continual pre-training setup resembling fine-tuning were addressed by clarifying that training is unsupervised next-token prediction over randomly repacked gold and distractor documents, with no access to questions or answers. Additional experimental details and ablations further alleviated these concerns.

Reviewer DGex: The role of cross-document attention versus increased compute was clarified through ablations showing that packing without cross-document attention does not yield gains. Data usage and training setup were also clarified.
**The paper does not yet provide direct mechanistic evidence explaining why cross-document attention improves reasoning, and normalized compute analyses are limited.**

Reviewer tn2x: Ambiguities in attention masking were resolved. Additional experiments with random global packing demonstrated that gains are not solely due to oracle packing. Clarifications were provided on metrics, variance across seeds, and reproducibility. **Benchmark scope remains limited, and compute normalization and statistical reporting could be further strengthened.**

**Reviewer Scores:**

All the three reviewers' may keep their scores.

---

### Decision · Program_Chairs · 2026-01-26

Reject